# Role of Flavonoids in the Prevention of AhR-Dependent Resistance During Treatment with BRAF Inhibitors

**DOI:** 10.3390/ijms21145025

**Published:** 2020-07-16

**Authors:** Héloïse M. Leclair, Nina Tardif, Anaïs Paris, Marie-Dominique Galibert, Sébastien Corre

**Affiliations:** 1Institut de Génétique et Développement de Rennes, University Rennes–UMR6290, F-35000 Rennes, France; heloiseleclair3@gmail.com (H.M.L.); nina.tardif@univ-rennes1.fr (N.T.); anais.paris@univ-rennes1.fr (A.P.); 2Department of Molecular Genetics and Genomics, Hospital University of Rennes, F-35000 Rennes, France

**Keywords:** melanoma, BRAF inhibitor, resistance, AhR, flavonoids

## Abstract

BRAF and MEK inhibitors (BRAFi and MEKi) are the standard of care for the treatment of metastatic melanoma in patients with BRAF^V600E^ mutations, greatly improving progression-free survival. However, the acquisition of resistance to BRAFi and MEKi remains a difficult clinical challenge, with limited therapeutic options available for these patients. Here, we investigated the therapeutic potential of natural flavonoids as specific AhR (Aryl hydrocarbon Receptor) transcription factor antagonists in combination with BRAFi. Experimental Design: Experiments were performed in vitro and in vivo with various human melanoma cell lines (mutated for BRAF^V600E^) sensitive or resistant to BRAFi. We evaluated the role of various flavonoids on cell sensitivity to BRAFi and their ability to counteract resistance and the invasive phenotype of melanoma. Results: Flavonoids were highly effective in potentiating BRAFi therapy in human melanoma cell lines by increasing sensitivity and delaying the pool of resistant cells that arise during treatment. As AhR antagonists, flavonoids counteracted a gene expression program associated with the acquisition of resistance and phenotype switching that leads to an invasive and EMT-like phenotype. Conclusions: The use of natural flavonoids opens new therapeutic opportunities for the treatment of patients with BRAF-resistant disease.

## 1. Introduction

The treatment of BRAF-mutant melanomas with MAP kinase (MAPK) pathway inhibitors is paradigmatic in precision cancer therapy but also highlights the problem of drug resistance, which limits the benefit to patients. MAPK inhibitors, such as BRAF inhibitors (e.g., vemurafenib, dabrafenib, and encorafenib), MEK inhibitors (e.g., cobimetinib, trametinib, and binimetinib), or their combination, are beneficial for most melanoma patients with tumor carrying activating V600E/K mutation in the BRAF oncogene [1]. However, they commonly fail to cure disease due to the development of resistance. Diverse oncogenic processes have been shown to be associated with acquired resistance, such as mutations in components of the MAPK pathway [2,3,4,5], reactivation of parallel signaling networks, such as PI3K/AKT [6,7,8], or the overexpression of various tyrosine kinase receptors (TKRs) (PDGFR, EGFR, NGFR) [2,9,10]. Melanoma cells are not only capable of rapidly adapting to therapies by acquiring de novo mutations, but they also tend to switch their molecular and cellular phenotype in an epithelial-to-mesenchymal transition (EMT)-like manner to bypass drug treatment. The most common such phenotypic changes are linked to the ratio of expression between the master transcription factor MITF and the TKR AXL [11,12,13,14]. The balanced expression of these two factors participate, among other actions, in differentiation/dedifferentiation, changes in proliferation rates and metabolic rewiring. Drug resistance can also arise through the selection of pre-existing subclones harboring cancer stem cell-like properties that are able to withstand drug treatment [15]. Indeed, drug-induced epigenetic reprogramming, initiated by the loss of SOX10-mediated differentiation [15], activates several transcription factors, including TEAD [16], allowing melanoma cells to survive. Recently, several studies identified a de-differentiation process associated with phenotype switching. During the acquisition of resistance, cells undergo a switch from a proliferative/pigmented phenotype to an invasive/dedifferentiated phenotype involved in BRAFi resistance, further highlighting the plasticity of melanoma cells during treatment [17,18,19].

We recently demonstrated that in melanoma, BRAFi constitute a new class of aryl hydrocarbon receptor (AhR) transcription factor ligands that promote sensitivity by triggering a transcriptomic program associated with cell differentiation. Conversely, AhR is constitutively activated in a subset of melanoma cells, promoting their dedifferentiation, associated with the expression of BRAFi-resistance genes. Under BRAFi pressure, this small subpopulation of AhR-activated and BRAFi-persister cells are enriched, leading to relapse [20]. We have also shown that the natural AhR antagonist resveratrol, used in combination with BRAFi, prevents the emergence of persister-resistant cells and thus delays tumor growth in vivo.

AhR is the only ligand-dependent transcription factor of the basic-helix-loop-helix (bHLH) Per–Arnt–Sim (PAS) family [21]. AhR is involved in many physiological processes [22,23], diseases, and types of cancer [24]. Several exogenous and endogenous ligands have been described to be AhR-agonists or antagonists [25]. Among them, we focus on the role of natural flavonoids to control AhR transcriptional activity and their function in melanoma treatment.

The role of natural flavonoids in cancer prevention has long been described for several types of cancer [26]. Dietary flavonoids are the most common polyphenols found in fruits, vegetables, flowers, chocolate, tea, wine, and other plant sources [27,28]. With more than 9000 members in this family, flavonoids can be divided into several subfamilies, including flavones, flavanols, isoflavones, flavonols, flavanones, and flavanonols, which share a basic chemical structure consisting of two benzene rings connected by a three-carbon bridge, forming a heterocycle (C6-C3-C6) [29]. Several studies have suggested that the dietary intake of flavonoids reduces the risk of developing certain types of cancer [30,31]. Several types of flavonoids have been shown to be antiproliferative and have preventive effects against various cancers [31]. Their chemopreventive effect is mediated by the induction of apoptosis, cell cycle arrest and the inhibition of metabolizing enzymes (notably cytochromes P450), reactive oxygen species formation, and angiogenesis [31].

Here, we focused on the role of a set of flavonoids in the control of the transcriptional activity of AhR [25,32,33] in the context of cutaneous melanoma (Appendix A), some of which have shown to inhibit the development, growth, and spreading of melanoma through their antioxidant properties (Appendix A).

In this study, we underscored the role of several flavonoids as AhR antagonists and showed their capacity to delay the acquisition of resistance to BRAFi.

## 2. Results

### 2.1. Flavonoids Are AhR Ligands That Control Its Transcriptional Activity

We and others previously reported that binding of the AhR PAS-B domain by its canonical ligands (TCDD, FICZ, BaP, etc.) at a position we arbitrarily named the α−pocket could be blocked by the binding of antagonists in the same α−pocket (CH-223191, resveratrol). We thus proposed the use of AhR inhibitors to prevent the canonical AhR activity associated with dedifferentiation and the resistance phenotype during BRAFi treatment of melanoma [20]. Flavonoids are small planar molecules that have been previously shown to activate or antagonize AhR transcriptional activity, as shown for resveratrol [32]. We further characterized flavonoids (Appendix A for their ability to bind and modulate AhR activity by conducting docking experiments using a AhR PAS-B model and flavonoid chemical structures recovered from the ZINC database (zinc.docking.org). Prediction of the interacting amino acids showed that, apart from naringenin (no binding), all tested flavonoids bind inside the α−pocket, similarly to canonical AhR ligands (TCDD, FICZ, BaP), (Figure 1a,b and Appendix A). However, unlike dioxin (TCDD) and benzo(a)pyrene (BaP), they all failed to activate the canonical AhR pathway, based on *CYP1A1* mRNA expression (Figure 1c), suggesting a role as an AhR antagonist. We thus tested their potential role as AhR antagonists. We performed XRE luciferase reporter assays in the presence of TCDD (10 nM) alone or in combination with the various flavonoids (10 µM) in the 501Mel human melanoma cell line (Figure 2a). Apart from naringenin, all flavonoids tested significantly inhibited the XRE-dependent luciferase activity induced by TCDD (Figure 2a). AhR competition assays performed with TCDD and increasing doses of flavonoids showed their activity (apart from naringenin) to be comparable to that of the prototypical AhR antagonist CH-223191 (CH) (1 to 10 µM flavonoids inhibited 50% of TCDD induction, Figure 2b). Measurement of *CYP1A1* mRNA expression levels in 501Mel cells after induction by TCDD (10 nM), alone or in combination with flavonoids (10 µM) (Figure 2c), and EROD enzymatic activity in MCF7 cells (Figure 2d) showed apigenin (Api), chrysin (Chr), and kaempferol (Kae) to be the most powerful antagonists of canonical AhR activity. Based on these results, we evaluated the role of these flavonoids on the melanoma phenotype during BRAFi treatment. Interestingly, as shown in Figure 1 and Figure 2, the flavonoids tested share similar structures but their antagonistic function on AhR is different. This suggest that the AhR-antagonist role of flavonoids must be linked to their differential ability to interact in the PAS-B domain of AhR.

### 2.2. Flavonoids Increase BRAFi Sensitivity and Diminish the Pool of Persister Cells

Before evaluating the biological impact of flavonoids in combination with BRAFi, we first analyzed their toxicity in several cell lines (501MelR and SKMel28R BRAFi-resistant melanoma cell lines and HaCat keratinocytes). Cells were treated for 48 h with increasing doses of flavonoids (0–100 µM) (Figure 3a). Treatment with flavonoids induced a significant decrease in cell density at a dose > 20 µM (Figure 3a) (IC50 values for each cell line are represented in the heatmap in Appendix A), corresponding to the induction of apoptosis and necrosis (Figure 3b). At lower doses, we did not observe morphological differences nor toxicity after treatment of melanoma cells with the different flavonoids. We then evaluated the impact of using flavonoids in combination with BRAFi on melanoma cell proliferation by quantifying the cell density of SKMel28R cells after 48 h of treatment with increasing doses of flavonoids (0 to 100 µM), alone or in combination with vemurafenib (Vem, IC50 dose = 5 µM. Flavonoids, such as Api and Chr, induced a significant decrease in cell density (Figure 3c). There was an additional effect in the presence of the BRAF inhibitors, making it possible to significantly reduce the pool of residual resistant cells (Figure 3c). We next evaluated the possibility that pre-treatment of melanoma cells with flavonoids may increase their sensitivity to BRAFi. SKMel28R and 501MelR cells were pretreated with flavonoids (10 µM) for 48 h and washed before treatment with 5 µM Vem for four days (every 2 days). This allowed comparison of the cell density after Vem treatment, alone or in combination with the flavonoids (Appendix A). Api significantly increased cell death induced by BRAFi (5 µM). Its effect was comparable to that observed with CH-223191 (Appendix A). 

Prolonging the pre-treatment of melanoma cells (SKMel28R) (at low doses = 5 µM) to 10 days significantly increased cell death, diminishing the pools of resistant cells (Appendix A) and improving BRAFi-sensitivity, with a significant decrease in the IC50 (Figure 3d). This effect was associated with a significant decrease in the expression of resistance genes in long-term treated cells (*AXL*, *NRP1*, *LPAR1*) (Figure 3e). We finally analyzed the ability of residual BRAFi-resistant cells to reproliferate while maintaining pressure with Vem. SKMel28R cells were treated with Vem alone (5 µM) or in combination with flavonoids (5 µM) every two days. Less than 20% of cells remained at this stage corresponding to residual resistant cells. We then analyzed the level of proliferation of these resistant cells under BRAFi pressure (Figure 3f). Interestingly, pretreatment with Api, Chr, Kae or CH prevented the proliferation of these residual resistant cells when compared to cells pretreated with Fis, RSV, Nar or with BRAFi alone, which were able to proliferate again. Although this diminution in cell proliferation is significant, it is impossible due to the absence of measurement points beyond 11 days to conclude as to the long-term efficacy of Api, Chr or Kae pretreatment.

Based on these results, we were able to identify Api, Chr, and Kae as powerful inhibitors, even at low concentrations, that potentiate the BRAFi-effect to kill melanoma cells by increasing cell death, as well as by preventing the induction of the expression of resistance genes (Figure 3e). The use of BRAFi combined with these flavonoids, which counteract canonical AhR activity, avoids the emergence of the BRAFi-resistant cells responsible for melanoma relapse. Despite the high similarity of the structures between the flavonoids (Figure 1b), their function on AhR and especially melanoma-cell survival under BRAFi treatment was significantly different. This may explain why the combination of flavonoids (#3 = Api + Chr + Fis, #2 = Api + Chr) with a comparable chemical structure did not potentiate their effect on BRAFi sensitivity.

### 2.3. Flavonoids Prevent Migration and Invasion of Resistant Melanoma Cells

Melanoma cells can acquire resistance to targeted therapies by switching from a proliferating to invasive state under the control of cellular plasticity [17,18]. These changes are associated with an aggressive phenotype similar to EMT, which favors metastatic spreading [34]. We recently showed the involvement of AhR transcription factor in the acquisition of resistance during melanoma treatment by BRAFi. Although AhR activation by BRAFi induces an expression program associated with the sensitive/differentiate state of melanoma cells, canonical activation of AhR switches cells to an expression program associated with the resistant/dedifferentiated/invasive state [20]. Here, we analyzed the impact of flavonoids on the melanoma phenotype. We first performed a clonogenic assay on SKMel28R cells as an in vitro cell survival assay to evaluate the ability of a single cell to grow into a colony after, or not, treatment with the various flavonoids. Melanoma cells were seeded into six-well plates at 200 cells per well. After adherence, the cells were exposed to flavonoid treatment (10 µM) and cultivated for 12 days. The clonogenic assays showed lower cell survival of SKMel28R cells in the presence of Api and Chr than the prototypical AhR antagonist (CH) and all other flavonoids (Figure 4a). We then performed wound-healing assays to evaluate the role of flavonoids on cell migration. Short-term treatment of SKMel28R melanoma cells with flavonoids (10 µM for 2 days) (especially Api, Chr, and Fis) significantly delayed cell migration (Figure 4b and Appendix A). This inhibitory effect was increased by long-term treatment of SKMel28R cells with Api, Chr, or Fis (5 µM for 10 days) (Figure 4c and Appendix A). Finally, we analyzed the effect of the various flavonoids on the invasive potential of SKMel28R melanoma cells using spheroid assays. Indeed, these BRAFi-resistant cells are highly invasive and characteristic of the metastatic phenotype. Both short-term (Figure 4d and Appendix A) and long-term treatment (Figure 4e and Appendix A) of the cells with AhR antagonist CH, but also Api or Chr, significantly reduced the invasive ability of SKMel28R cells. The combination of the flavonoids Api + Chr (#2) and Api + Chr + Fis (#3) had the same effect as the flavonoids alone on cell migration (Figure 4b,c and Appendix A) and invasion of SKMel28R melanoma cells. In conclusion, the flavonoids Api and Chr appear to be the most potent to counteract cell proliferation, migration, and invasion of BRAFi resistant melanoma cells in vitro (Appendix A).

### 2.4. Treatment of SKMel28-Resistant Cell Lines by Apigenin Leads to a Decrease in the Expression of Genes Associated with the Resistance Phenotype

We next performed RNAseq analysis of SKMel28R cells treated, or not, with Api (1 µM for 48 h) to decipher its role in counteracting the resistance phenotype and sensitizing melanoma cell lines to BRAFi. After treatment, the expression of 440 differentially expressed genes (DEGs) were significantly enriched or diminished (Fold Change FC < 1.5; False Discovery Rate FDR < 0.01) (Figure 5a). Functional annotation (human KEGG pathways, Enrichr analysis (https://amp.pharm.mssm.edu/Enrichr)) showed the DEGs to be particularly involved in interactions with the extracellular matrix (ECM), focal adhesion, or tight junctions (Figure 5b). These pathways are particularly important for spreading of the tumor into its proximal environment. Other genes were associated with the activation of cellular pathways (PI3K, p53, FOXO) previously shown to be associated with the resistance/invasive phenotype in melanoma (Figure 5b) [35,36,37,38]. We then performed gene-set enrichment analysis (GSEA) (http://software.broadinstitute.org/cancer/software/gsea) by comparing Vem-sensitive or -resistant melanoma cell lines from the Cancer Cell Line Encyclopedia (CCLE) RNA seq dataset [39] to identify from among these DEGs (apigenin vs. CTR) those that are significantly enriched in the resistant state (Figure 5c). We selected the top 75 significantly enriched genes in resistant cell lines to analyze their expression in SKMel28R melanoma cells treated, or not, with Api (Figure 5d, left) and compared it to that of SKMel28S melanoma cells. The expression of 65% (49/75) of the genes associated with the resistance phenotype was significantly lower in both SKMel28R cells treated with Api and SKMel28S cells (BRAFi-sensitive) (Figure 5d, left). In addition, the genes overexpressed in SKMel28R cells and downregulated by Api correlated with an invasive phenotype of melanoma cells, as confirmed by gene-expression analysis in melanoma cell lines from the dataset of Verfaillie et al. (GSE60664 [16]), depending on their proliferative (green) or invasive state (red) (Figure 5d, right). Furthermore, we found that the acquisition of BRAFi resistance correlates with a progressive increase in the expression of selected genes from the parental sensitive cell lines using the dataset of an additional melanoma cell line (M229) [40,41], with one pool of genes overexpressed during the persister latency phase (DTP: non-proliferative; DDTP: proliferative) and another overexpressed later during the resistance and relapse phase (SDR: single drug or DDR: double drug) (Appendix A).

### 2.5. Apigenin Increases BRAFi Sensitivity of Melanoma Xenografts in a Chick Embryo Model

We next investigated the ability of Api to sensitize melanoma cells to BRAFi in vivo using the chick choroallantoic membrane (CAM) assay, which has been previously used to study tumor growth [42]. SKMel28R melanoma cells (1.10^6^) were grafted onto the CAM of each egg (E9). The eggs were randomized into groups for everyday treatment (from E10 to E19) with either vehicle, Vem alone (1 mg/kg), Api alone (1.65 mg/kg), or in combination (Figure 6a). The dose of Api used is significantly lower than the doses used in other studies (mouse models: 20–50 mg/kg [43,44]; human clinical trials: up to 500mg capsule 3 times daily of chamomile pharmaceutical grade [45,46]). This dose did not induce toxicity in the egg model. Tumor growth was measured on day 19 (E19). Vem alone induced a significant decrease in tumor volume relative to the vehicle (*p* = 0.0108) (Figure 6b), whereas Api alone did not decrease tumor size. Importantly, the Vem/Api combination induced significantly greater diminution of the tumor than Vem alone. Overall, these data support that co-treatment with Api, even at low concentrations, sensitizes melanoma cells to BRAFi treatment in vivo (Figure 6b).

## 3. Discussion

Recent therapeutic advances in melanoma treatment have greatly improved patient outcomes. Combined BRAF and MEK inhibition has increased median patient free survival PFS [47], but patient overall survival could still be improved by preventing the emergence of drug resistance of late stage metastatic melanoma. Recent studies established that melanoma is a heterogeneous tumor with a very high cellular plasticity, contributing to the acquisition of resistance [17,18,48,49]. We recently involved the canonical activation of AhR transcription factor in the acquisition of resistance during BRAFi treatment of melanoma [20]. Interestingly, we could counteract canonical AhR activity with specific antagonists, such as the natural flavonoid resveratrol to decrease the pool of BRAFi resistant/persister cells, and thus delay tumor relapse.

In line, the present study aimed to test new combination treatments with BRAF inhibitors that could act synergistically and thus prevent or delay resistance. We mainly focused on the role of natural flavonoids to block canonical AhR activity. Flavonoids are polyphenols found in numerous edible plant species, such as fruits, vegetables, grains, roots, flowers, tea, and wine. Data obtained from preclinical and clinical studies suggest that some flavonoids endorse chemo-preventive activity and are cytotoxic against various cancers, notably melanoma [26]. These flavonoids have already been studied in cancer for their capacity to induce apoptosis, inhibit cell proliferation and invasion and for their antioxidant properties [50].

Here, we were particularly interested in the role of flavonoids as specific AhR antagonists [32,51,52]. We show that, despite the similar chemical structures of various flavonoids, their role in controlling the transcriptional activity of AhR can be very different. Flavonoids interact with the PAS-B binding domain differently, leading to distinct antagonist capacity. [51]. Among the flavonoids tested, Api was the most powerful to counteract BRAFi resistance in metastatic melanoma by antagonizing AhR activity. Indeed, Api appeared to be highly effective, even at low concentrations, in sensitizing melanoma cells in vitro and in vivo to BRAFi treatment. Treatment with Api effectively counteracts the acquisition of an invasive melanoma phenotype and significantly delays relapses associated with resistance to BRAFi. The beneficial effect of Api on tumor progression was mediated by inhibiting the transcriptional activity of AhR to regulate the expression of genes associated with BRAFi-resistance. Among them we identify *BIRC3, SLIT2, GBP2,* and *AFAP1* (Figure 5d) to be involved in the acquisition of BRAFi resistance in melanoma cell lines [53].

Apigenin (4′,5,7-trihydroxyflavone), one of the most widespread flavonoids in plants has been under tight investigation for its anti-cancer activities and its low toxicity [54]. Api was reported to impact in vitro and in vivo the biology of various human cancers by triggering cell apoptosis [55] and autophagy, inducing cell-cycle arrest [55,56], inhibiting cell migration and invasion [55], stimulating the immune response [57], and suppressing cancer stem cells (CSCs) [58,59]. Api has thus shown broad anticancer effects in various types of cancers, including colorectal, breast, liver, lung, prostate cancers, melanoma and osteosarcoma [55,60,61,62,63]. However, Api had only moderate anti-cancer activity at physiological concentrations due to its very low solubility in water (1.35 mg/mL) and stability, limiting its bioavailability [31,64,65]. Several delivery systems (liposomes, polymeric micelles, nanosuspension) have been used to improve Api solubility. In particular, nanosized-drug delivery systems (nanocapsules) are promising tools to prolong the pharmacological activity of apigenin [66,67].

We propose to use Api as a nutraceutical formulation in combination with conventional anti-melanoma therapies [68,69], especially since the use of the beneficial properties of natural products corresponds to widely present habits of food supplements notably in dermatology [70]. In addition, several phase II clinical trials using flavonoids have shown their efficacy in different cancers (colorectal, breast and prostate) [71,72,73]. Thus co-treatment could be a reasonable way to enhance BRAFi/MEKi anti-melanoma activity, as demonstrated previously using combination therapies with both apigenin and chemodrugs [54].

Together, this study underscores the role of Api as an AhR antagonist, demonstrating its capacity to delay the acquisition of resistance to BRAFi. These results have important clinical impact and may change first line treatment of BRAF V600-mutated metastatic melanoma.

## 4. Methods

### 4.1. Reagents (AhR Ligands)

2,3,7,8-TCDD (TCDD, 48599) was purchased from Sigma Aldrich (US), and vemurafenib (Vem, PLX4032, RG7204), CH-223191 (S7711), and all the various natural flavonoids (Appendix A) from Selleckchem (US).

### 4.2. Cell Lines and Culture Conditions 

Human mammary MCF7 epithelial cells and HaCat keratinocytes were cultured in humidified air (37 °C, 5% CO_2_) in Dulbecco’s modified Eagle’s medium with 4500 mg/L D-glucose and 110 mg/L sodium pyruvate, supplemented with 10% fetal bovine serum (FBS, Eurobio Scientific, FR) and 1% penicillin–streptomycin antibiotics (Gibco, Invitrogen, Paisley, UK). The melanoma cell lines (501Mel, SKMel28) were grown in humidified air (37 °C, 5% CO_2_) in RPMI-1640 medium (Gibco, Paisley, UK, Invitrogen,) supplemented with 10% FBS. SKMel28 (S + R) cells were obtained from the laboratory of J.C Marine at the VIB (Vlaams Instituut voor Biotechnologie, Center for Cancer Biology, Leuven, Belgium. 501Mel cells (S) were obtained from the ATCC and 501Mel BRAFi resistant cells (R) were obtained after three months of treatment with Vem (1 µM every 2 days). All cell lines were routinely tested for mycoplasma contamination (Mycoplasma Contamination Detection Kit, rep-pt1, InvivoGen, Toulouse, FR).

### 4.3. Molecular Modelling

Docking experiments were performed using AutoDock4.2 (free open tool, http://autodock.scripps.edu.) [74]. A multiple alignment between the sequences of PAS-B mAhR (residues 278–384) and PAS-B HIF-2α was generated according to the sequence alignment suggested by Pandini et al. [75]. The homology model of PAS-B mAhR was constructed using the crystal structures of the heterodimer complex of PAS-B HIF-2α (pdb code: 3f1p, 3f1o, 3f1n) and Prime v.2.1. docking experiments were carried out between the AhR PAS-B model and the chemical structures of various AhR ligands, BRAFi, and flavonoids recovered from the ZINC database (zinc.docking.org).

### 4.4. Evaluation of Cell Density

Cell density was assessed using a methylene blue colorimetric assay [76]. Briefly, cells were fixed for at least 30 min in 95% ethanol. Following ethanol removal, the fixed cells were dried and stained for 30 min with 1% methylene blue dye in borate buffer. After four washes with tap water, 100 µL 0.1 N HCl was added to each well. The plates were then analyzed with a spectrophotometer at 620 nm.

### 4.5. Wound-Healing Migration Assay

Briefly, SKmel28 cells were grown until confluency in two-well silicone inserts (Ibidi^®^, Gräfelfing, Germany) were placed in 12-well tissue culture dishes. The cell culture insert was removed after one day. Afterwards, the plates were washed with PBS and incubated at 37 °C in fresh RPMI-1640 medium (Gibco BRL, Invitrogen, Paisley, UK) supplemented with 10% FBS (Eurobio Scientific, Evry, France) and 1% penicillin–streptomycin antibiotics (Gibco, Invitrogen) in the presence of a vehicle (DMSO) or the indicated concentrations of flavonoids. The wound was photographed at 5× magnification with an Axio Vert.A1 inverted microscope (Carl Zeiss, Vision Aalen, Germany). Wound closure was determined by measuring the distance between the edges of the wound at 0 and 15 h using ImageJ (Fiji 1.0). The distance migrated by the cells was quantified as follows: D = (size of the wound at *t* = 0 h – size of the wound at *t* = 15 h).

### 4.6. Spheroid-Formation Assay

The spheroid-formation assay was performed as previously described. Briefly, SKMel28 cells (7000 cells/mL) were plated in 24-well plate coated with 1.5% agarose in complete RPMI medium and concentrated in the center by circular agitation. After two days, spheroids were recovered for inclusion in an extracellular matrix of collagen (100 µL) (final concentration = 2 mg/mL in buffer (acetic acid 0.01N; neutralization buffer: 33 mM Hepes pH7.4, 0.37% sodium bicarbonate, 0.03 N NaOH; MEM 1X) in 24-well plates coated with 1.5% agarose. Spheroids were maintained in complete medium, with or without flavonoids, and images of the spheroids were acquired over several days (0 to 4 days) at 5× magnification with an Axio Vert.A1 inverted microscope (Carl Zeiss). Invasion capacity was evaluated by determining the ratio between the maximum and initial diameter of the spheroid.

### 4.7. Colony-Formation Assay

Melanoma cells were seeded in six-well plates at 200 cells per well. After complete cell adherence, the cells were exposed to flavonoid treatment (10 µM) and cultivated for 12 days. The medium with the flavonoids was changed every two days. At the end of the experiment, colonies were stained with 0.5% crystal violet (50% H_2_O, 50% methanol) and manually counted. Each measurement was performed in triplicate and three independent experiments were performed.

### 4.8. Apoptosis and Necrosis Assays

At the end of treatment, cells were stained by adding 25 µL of a dye mixture containing Hoechst33342 (5 µg/mL) propidium iodide (4 µg/mL), and YO-PRO^®^-1 (0.8 µM) directly into 100 µL of culture media and incubated at 37 °C for 30 min. Cells were imaged, analyzed, and counted using an ArrayScan™ VTI High-Content System (ThermoFisher Scientific, Courtaboeuf, France). The apoptotic (green) and necrotic (red) cells are expressed relative to the number of Hoechst-positive cells (blue).

### 4.9. Ethoxyresorufin O-Deethylase Activity Assay

Ethoxyresorufin O-deethylase (EROD) activity, corresponding to the O-deethylation of ethoxyresorufin and mainly catalyzed by the CYP1A1 enzyme in living MCF7 cells, was measured as described previously [77].

### 4.10. Luciferase Activity

Reverse transfection of 501Mel cells was performed in 48-well plates simultaneously to cells plated with the pGL3-XRE3-FL [77] construct carrying firefly luciferase, using the Lipofectamine 2000 transfection reagent^®^ according to the manufacturer’s instructions (ThermoFisher Scientific, Waltham, MA, US). Twenty-four hours after transfection, cells that were transfected with the same luciferase reporter were exposed to TCDD (as positive control of transfection of all the cells and of AhR activity), with or without flavonoids, for 16 h. Luciferase assays were then performed with a Promega kit according to the manufacturer’s instructions. Data are expressed in arbitrary units relative to the value of the luciferase activity levels found in DMSO-exposed cells, arbitrarily set to one arbitrary unit (a.u.). Firefly luciferase activity was normalized to protein content using the Bicinchoninic Acid Kit from Sigma-Aldrich^®^ and measured using a luminometer (SAFAS Xenius XL, SAFAS Monaco).

### 4.11. RNA Extraction and RT-qPCR Expression 

Experiments were performed as previously described [78]. Primers used for the RT-qPCR experiments are shown in Appendix A.

### 4.12. RNA-Seq

Total RNA was extracted from BRAFi-sensitive or resistant SK28 cells, in the presence or absence of Api, using the miRVana kit (ThermoFisher Scientific, Waltham, US). RNAseq was performed in collaboration with Novogene. Libraries were generated from 500 ng total RNA using the Truseq Stranded mRNA kit (Illumina). Libraries were then quantified using the KAPA library quantification kit (Kapa biosystems, Sigma-Aldrich, St Louis, US) and pooled. A portion (0.5 nM) of this pool was loaded onto a high output flowcell and sequenced on a NextSeq500 platform (Illumina) with 2 × 75 bp paired-end chemistry within two runs. Reads were aligned to human genome release hg19 using STAR v2.4.0a with default parameters. Gene quantification was then performed using featureCounts release subread-1.5.0-p3-Linux-x86_64 with “--primary -g gene_name -p -s 1” options. Quality control of the RNA-Seq count data was performed using in-house R scripts. Normalization and statistical analysis were performed using the Bioconductor package DESeq2. *p*-values were adjusted for multiple testing using the Benjamini–Hochberg procedure, which controls the false discovery rate (FDR). Differentially expressed genes were selected based on an adjusted *p*-value < 0.05. RNAseq data are available on request. The functional annotation of differentially expressed genes was performed using the Enrichr webtool (https://amp.pharm.mssm.edu/Enrichr).

### 4.13. Data Mining

Expression heatmaps of differentially expressed genes between samples were obtained based log2 fold changes using the ComplexHeatmap 2.0.0 [79] package in R/Bioconductor. Cluster-specific gene rankings were obtained by contrasting each sample with the rest of the samples. Cell density curves for the available melanoma cell lines were established using GraphPad PRISM 8.0 ® to establish IC50 dependently of the different treatments. The raw data count matrix from RNA seq data were obtained in the GEO database for previous experiments on melanoma cell (proliferative or invasive) GSE60664 (https://www.ncbi.nlm.nih.gov/gds/?term=GSE60664) [16] and from the BRAFi- or BRAFi+MEKi- resistant versions of the cell line GSE75299 (https://www.ncbi.nlm.nih.gov/gds/?term=GSE752099) [41].

### 4.14. Tumor Grafts in the Chicken CAM Model

All elements of this assay (amplification of tumor cell lines, incubation of eggs, tumor grafting on the chicken embryo chorioallantoic membrane (CAM), embryo survival analysis, evaluation of tumor growth) were carried out at INOVOTION SAS (Biopolis, 5 avenue du Grand Sablon, La Tronche, 38700, France).

#### 4.14.1. Preparation of Chicken Embryos

Fertilized White Leghorn eggs were incubated at 37.5 °C at 50% relative humidity for nine days. At that time (E9), the CAM was exposed by drilling a small hole through the eggshell into the air sac and a 1-cm^2^ window was cut in the eggshell above the CAM. In the absence of toxicity of the treatment, at least 15 eggs are used for each group for data analysis.

#### 4.14.2. Amplification and Grafting of Tumor Cells

SKMel28R (vemurafenib resistant) tumor cells were cultivated in RPMI medium supplemented with 10% FBS, 1% pyruvate, and 1% penicillin–streptomycin. On day E9, cells were detached with trypsin, washed with complete medium, and suspended in graft medium. An inoculum of 1.10^6^ cells was added to the CAM of each egg (E9). The eggs were then randomized into groups.

#### 4.14.3. Treatments

On day E10, tumors begin to be detectable. The treatments corresponding to different groups are detailed in Figure 6A.

#### 4.14.4. Quantitative Evaluation of Tumor Growth

On day E18, the upper portion of the CAM (with tumor) was removed, washed with PBS buffer, and directly transferred to paraformaldehyde (fixation for 48 h). Then, the tumors were carefully cut away from the normal CAM tissue and weighed. A one-way ANOVA analysis with post-tests was performed on the data. 

### 4.15. Statistics

Data are presented as the means ± s.d. unless otherwise specified, and the differences were considered significant for *p* < 0.05. Comparisons between groups normalized to a control were carried out using a two-tailed *t*-test with the Holm–Sidak multiple comparisons test when more than two groups were compared to the same control condition. Overall survival was estimated using the Kaplan–Meier method. Univariate analysis using the Cox regression model was performed to estimate hazard ratios (HR) and 95% confidence intervals (CI). All statistical analyses were performed using Prism 7 software (GraphPad, La Jolla, CA, USA).

### 4.16. Data Availability

The datasets generated and/or analyzed during the current study are available from the corresponding author on reasonable request.

## Figures and Tables

**Figure 1 ijms-21-05025-f001:**
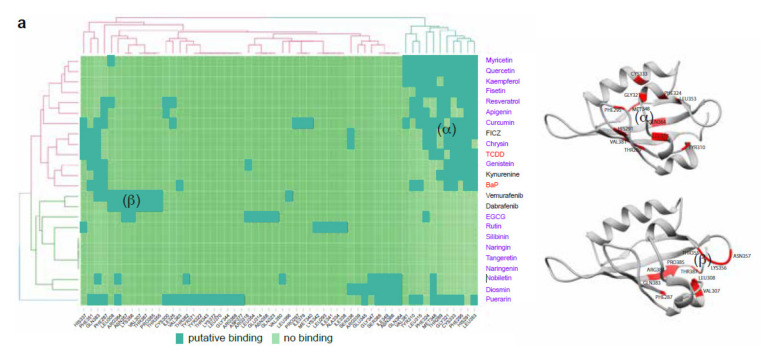
Flavonoids are potential AhR ligands. (**a**) Left: Heat map showing hierarchical clustering of AhR ligands or various flavonoids and their putative interactions with amino-acids of the PAS B domain of AhR. Several flavonoids cluster with canonical AhR ligands, such as TCDD, FICZ, BaP, and kynurenine, whereas BRAFis cluster together in a different position of the PAS B domain. Right: 3D representation of the PAS-B domain of AhR. Amino acids of the α (top) and β (bottom) binding pockets are highlighted in red. (**b**) Chemical structures and proposed binding mode of the natural flavonoids apigenin, chrysin, fisetin, kaempferol, resveratrol, and silibinin to AhR PAS-B ligand-binding domain homology model. The free binding energy is reported in Appendix A. The two predictive ligand-binding pockets are indicated by (α) or (β) (**c**) *CYP1A1* expression in 501Mel cells exposed to vehicle, flavonoids (1 µM), BRAFi: vemurafenib (Vem) and dabrafenib (Dab) (1 µM), or AhR ligands (TCDD) (10 nM) (*n* = 2) for 48 h.

**Figure 2 ijms-21-05025-f002:**
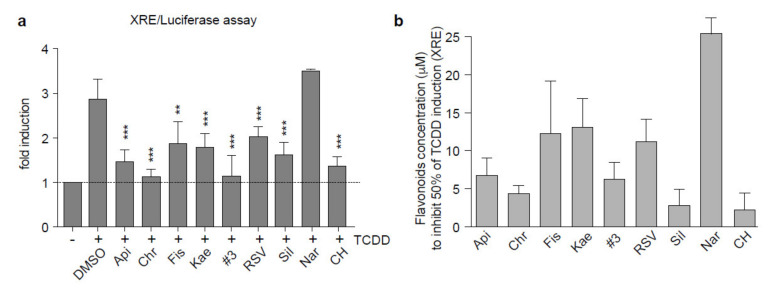
Flavonoids act as AhR antagonists against its canonical activity. (**a**–**d**) Flavonoids antagonize the canonical AhR activity induced by TCDD (dioxin). (**a**) Evaluation of AhR transcriptional activity related to AhR/ARNT binding sites (XRE) using p3XRE-luciferase constructs. 501Mel cells were exposed to 10 nM TCDD alone or in combination with flavonoids 10 µM (pretreated for 2 h) for 6 h (*n* = 3). (**b**) 501Mel cells were transfected with the p3XRE-luciferase construct and induced simultaneously with TCDD (10 nM) and increasing doses of flavonoids or CH-223191 to compete with AhR agonist. The histogram shows the required concentration of flavonoids to inhibit 50% of the XRE luciferase activity induced by TCDD (*n* = 3). (**c**) Flavonoids prevent the induction of *CYP1A1* mRNA expression by TCDD. 501Mel cells, pretreated or not with flavonoids (10 µM, 2 h), were incubated or not with 10 nM TCDD 15 h (*n* = 3). (**d**) MCF-7 cells were either untreated or treated with 10 nM TCDD or 10 µM flavonoids for 6 h. The ability of the flavonoids to prevent TCDD-induced EROD activity was measured (*n* = 3). Data correspond to the mean +/− s.d. of three independent experiments. Statistical analysis was performed using an unpaired t-test (PRISM8.0^®^) * *p* < 0.05, ** *p* < 0.01, *** *p* < 0.001. # 3 = Mix of apigenin, chrysin, and fisetin (10 µM final concentration).

**Figure 3 ijms-21-05025-f003:**
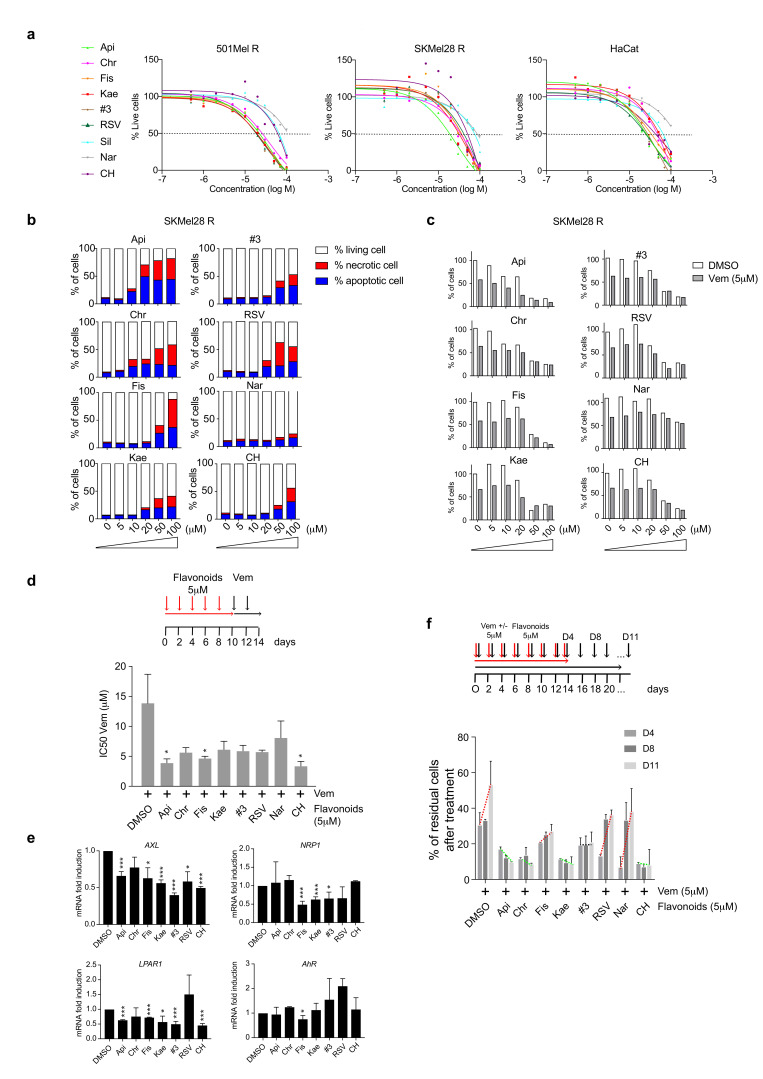
Flavonoids sensitize melanoma cells during BRAFi treatment (**a**) 501MelR, SKMel28R (BRAFi-resistant), and HaCat keratinocyte cells were treated for 48 h with increasing concentrations of flavonoids (0 to 100 µM). Cell density was evaluated by methylene-blue staining followed by quantification at 620 nm (*n* = 2). (**b**) SKMel28R (BRAFi resistant) cells were treated for 48 h with increasing concentrations of flavonoids (0 to 100 µM). At the end of the treatment, cells were stained with Hoechst33342, propidium iodide, and YO-PRO^®^-1 to evaluate the percentage of apoptotic (represented in red) and necrotic (represented in blue) cells relative to the number of Hoechst-positive cells by microscopy. (**c**) SKMel28R (BRAFi resistant) cells were treated, or not, with vemurafenib (Vem, IC50 dose = 5 µM) for 48 h in combination with increasing concentrations of flavonoids (0 to 100 µM). At the end of treatment, cells were stained with Hoechst33342 to evaluate the percentage of live cells. (**d**) SKMel28R (BRAFi resistant) cells were pretreated with flavonoids (5 µM) for 10 days and washed before treatment with increasing concentrations of vemurafenib for four days (every 2 days) to establish the IC50 for BRAFi alone relative to that of co-treatment with flavonoids. Values, calculated using GraphPad PRISM, are presented as the mean ± sem IC50 of Vem for control cells (DMSO) or after treatment with the various flavonoids (*n* = 3); unpaired *t*-tests with the Sidak–Bonferroni method. (**e**) Level of mRNA expression for resistant genes (*AXL, NRP1, LPAR1*) and *AhR* measured by RT-qPCR (*n* = 3) in SKMel28R (BRAFi resistant) cells treated with flavonoids (5 µM) for 10 days (Appendix A). (**f**) SKMel28R (BRAFi resistant) cells were treated or not with vemurafenib (IC50 dose = 5 µM, black arrows) for 14 days, every two days, in combination or not with flavonoids (5 µM, red arrows). At the end of treatment, the cell density was evaluated by optical microscopy and quantified by methylene-blue staining, followed by quantification at 620 nm (*n* = 2) at various times (4 to 11 days) to evaluate the ability of melanoma cells to relapse under BRAFi treatment. # 3 = Mix of apigenin, chrysin and fisetin (10 µM final concentration). Statistical analysis was performed using an unpaired t-test (PRISM8.0^®^) * *p* < 0.05, ** *p* < 0.01, *** *p* < 0.001.

**Figure 4 ijms-21-05025-f004:**
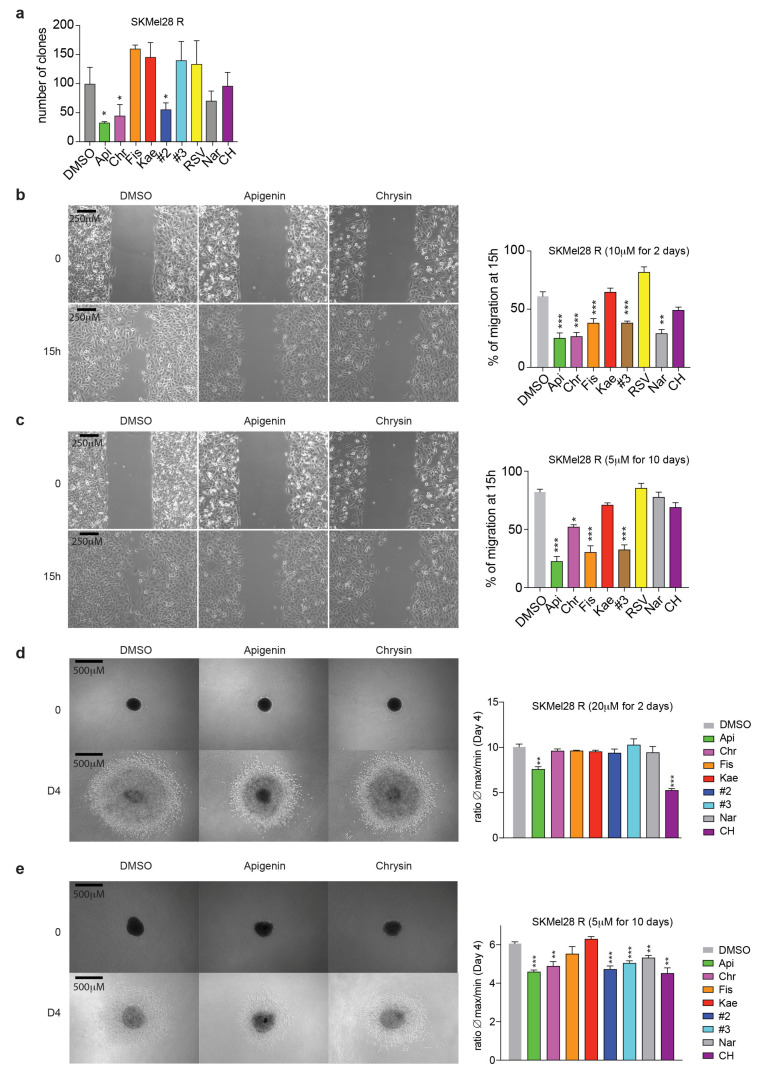
Flavonoids prevent the migration and invasion of BRAFi-resistant melanoma cells. (**a**) Clonogenic assays were performed to evaluate proliferation using SKMel28R cells exposed to flavonoid treatment (10 µM) and cultivated for 12 days. The histogram shows the mean number ± sem. of colonies (*n* = 3). (**b**,**c**) Wound-healing assays were performed using IBIDI^®^ chambers to evaluate the role of flavonoids on cell migration. Images of the wound were acquired at a 5× magnification with an Axio Vert.A1 inverted microscope (Carl Zeiss). The histogram shows the mean ± sem; wound closure determined by measuring the distance between the edges of the wound at time 0 and 15 h (*n* = 3–6), unpaired *t*-tests with the Sidak–Bonferroni method. SKMel28R cells were treated with 10 µM flavonoids. Scale bar 250 µm. (**b**) for two days (short term) or 5 µM flavonoids (**c**) for 10 days (long term) before performing the migration assay (0 to 15 h) in the absence of treatment. (**d**,**e**). Spheroid formation assays were performed in BRAFi-resistant SKMel28 cells to evaluate the role of flavonoids on cell migration. Images were acquired at days 0 and 4 after spheroid implantation into a collagen matrix. Each histogram shows the mean ± sem; unpaired *t*-tests with the Sidak–Bonferroni method. Scale bar 500 µm. (**d**) SKMel28R cells were treated with 20 µM flavonoids (**d**) for two days (short term) or 5 µM flavonoids (**e**) for 10 days (long term) before spheroid formation and inclusion in an extracellular matrix of collagen (long term). # 2 = Mix of apigenin and chrysin; # 3 = Mix of apigenin, chrysin and fisetin. Statistical analysis was performed using an unpaired *t*-test (PRISM8.0^®^) * *p* < 0.05, ** *p* < 0.01, *** *p* < 0.001.

**Figure 5 ijms-21-05025-f005:**
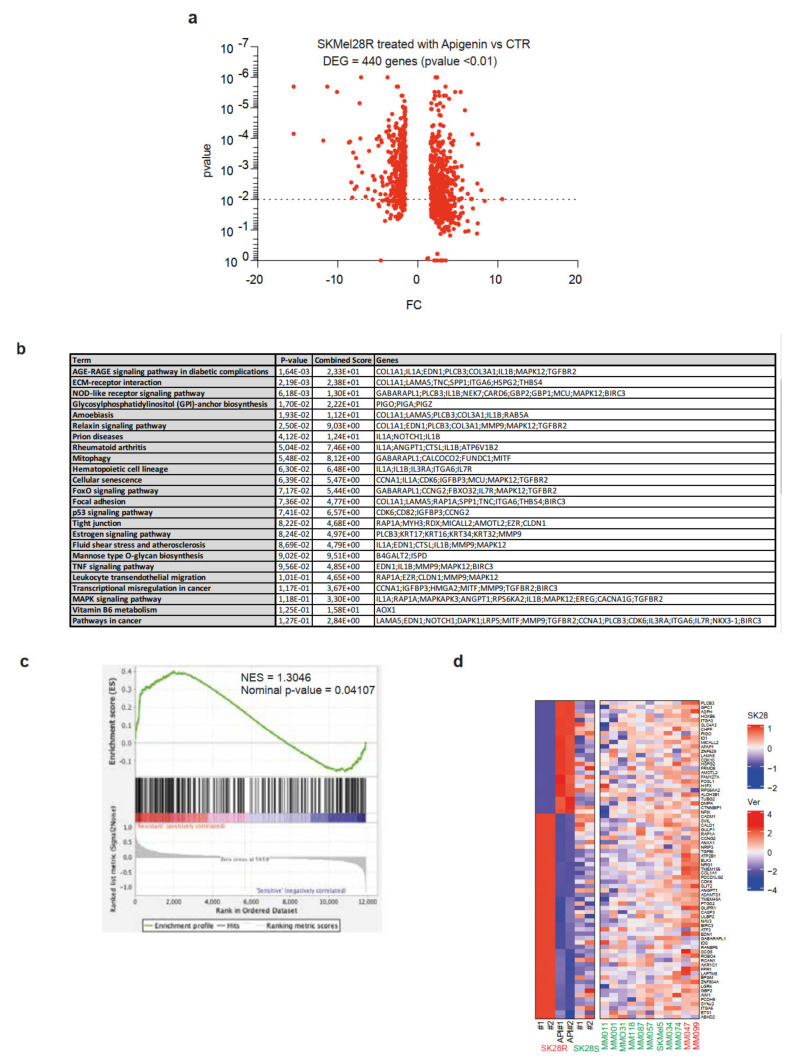
Apigenin treatment decreases the expression of genes associated with resistance in melanoma. (**a**) Volcano plot combining the magnitude of the fold change (ratio of expression between resistant and sensitive SK28 cell lines) and the *p*-values for differentially expressed genes between SKMel28R cells treated with DMSO or apigenin for 48 h (1 µM) (FC > 1.5; FDR < 0.01). (**b**) Human KEGG pathway enrichment of differentially expressed genes (DEGs) between SKMel28R treated with DMSO and SKMel28R treated with apigenin (Enrichr webtool, https://amp.pharm.mssm.edu/Enrichr). (**c**) Gene-set enrichment analysis (GSEA), showing that some DEGs between SKMel28R treated with DMSO and SKMel28R treated with apigenin were significantly enriched in BRAFi-resistant vs. sensitive melanoma cell lines from the Cancer Cell Line Encyclopedia RNAseq dataset (GSE36134 [39]). (**d**) Expression heatmap for the top genes enriched (*n* = 75) in BRAFi-resistant cell lines (SKMel28R) treated with DMSO or apigenin, SKMel28S (left panel), and the melanoma cell lines dataset from the GEO dataset (GSE60664 [16]), depending on their proliferative (green) or invasive states (red) (right panel). Genes and clusters with similar expression profiles across the cohort are placed close to each other on the grid.

**Figure 6 ijms-21-05025-f006:**
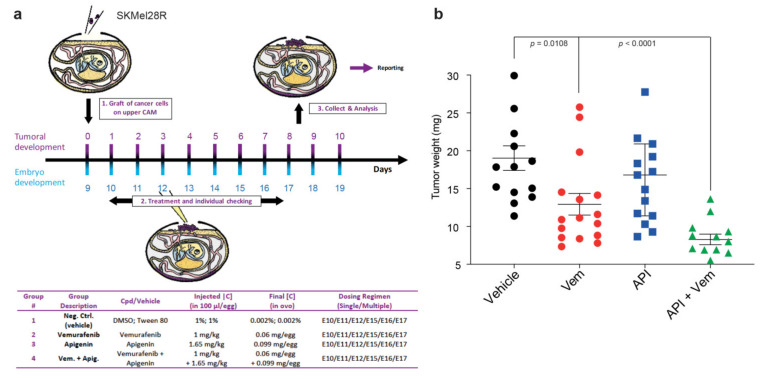
Apigenin increases tumor shrinkage of resistant melanoma cells treated with Vemurafenib. (**a**) Experimental protocol of grafting SKMel28R in the CAM (chick chorioallantoic membrane) of chicken embryos and treatment with vemurafenib alone or in combination with apigenin. In the absence of toxicity of the treatment, at least 12 eggs were used for each group for data analysis. An inoculum of 1.10^6^ cells (SKMel28R) was placed onto the CAM of each egg (E9). Then, the eggs were randomized into groups. On day E10, tumors begin to be detectable. Treatments corresponding to different groups are detailed. (**b**) Tumor weight (mg) nine days after daily treatment with vehicle (*n* = 13), vemurafenib (1 mg/kg) (*n* = 16), apigenin (1.65 mg/kg) (*n* = 14), or in vemurafenib + apigenin (1 mg/kg and 1.65 mg/kg) (*n* = 12) combined. Values correspond to the mean +/− sem. Unpaired *t*-tests with the Holm–Sidak correction method were performed to compare treatments.

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
