# Peer review of "Role of Flavonoids in the Prevention of AhR-Dependent Resistance During Treatment with BRAF Inhibitors"

_ijms, 2020, doi:10.3390/ijms21145025_

Round 1

Reviewer 1 Report

The authors in the manuscript entitled “Role of flavonoids in the prevention of AhR-dependent resistance during treatment with BRAF inhibitors” have investigated the therapeutic potential of natural flavonoids as a combination therapy with BRAFi in resistant melanoma cell lines. Many of the flavonoids examined acted as AhR antagonists. Using in vitro and in vivo experiments, the authors showed that the flavonoids could synergize with BRAFi treatment to potentiate their effect. The best combo treatment of apigenin + Vemurafenib showed reduction in tumor weight in vivo when resistant melanoma cells were grafted onto a chicken embryo. The work presented in the manuscript is quite interesting as it describes use of natural sources to combat resistant cancers. The manuscript is well written and the results are well described. The data supports the author’s conclusions. Minor comments and questions:

Since with the combo treatment for eg in Fig 3f, the resistant persister cells are not completely destroyed even at the high combination of Api+Vem, there is a possibility of relapse. What are the author’s comments on that? Also, would the author’s think that combo treatment throughout the course of treatment be more ideal option?

Do the authors plan to optimize the Api molecule to lower the IC50? As the authors presented the data using one in vivo model, do the authors expect potential issue of toxicity with the use of such high concentration of Api in combination with BRAFi, if the combination treatment gets pursued into the clinic?

There are a few typos and missing numbers. I quick read through should fix that.

Author Response

The authors in the manuscript entitled “Role of flavonoids in the prevention of AhR-dependent resistance during treatment with BRAF inhibitors” have investigated the therapeutic potential of natural flavonoids as a combination therapy with BRAFi in resistant melanoma cell lines. Many of the flavonoids examined acted as AhR antagonists. Using in vitro and in vivo experiments, the authors showed that the flavonoids could synergize with BRAFi treatment to potentiate their effect. The best combo treatment of apigenin + Vemurafenib showed reduction in tumor weight in vivo when resistant melanoma cells were grafted onto a chicken embryo. The work presented in the manuscript is quite interesting as it describes use of natural sources to combat resistant cancers. The manuscript is well written and the results are well described. The data supports the author’s conclusions. Minor comments and questions:

Since with the combo treatment for eg in Fig 3f, the resistant persister cells are not completely destroyed even at the high combination of Api+Vem, there is a possibility of relapse. What are the author’s comments on that? Also, would the author’s think that combo treatment throughout the course of treatment be more ideal option?

We thank the reviewer for this comment. As indicated in Fig. 3F, after pretreatment of the cells with Api, a small percentage of residual cells remains (<20%). However, 11 days after withdrawal of apigenin, there is no restart of the proliferation of resistant cells. But rather a decreasing curve. It is impossible due to the absence of measurement points beyond 11 days to conclude as to the long-term efficacy of the apigenin pretreatment. As judiciously proposed, the maintenance of the therapeutic combination as suggested by the in vivo experiments in Fig. 6 makes it possible to maintain the pressure on the resistant cells.

Do the authors plan to optimize the Api molecule to lower the IC50? As the authors presented the data using one in vivo model, do the authors expect potential issue of toxicity with the use of such high concentration of Api in combination with BRAFi, if the combination treatment gets pursued into the clinic?

Preliminary results obtained in vivo focus on the therapeutic combination between apigenin and BRAFi to slow relapses. The doses of apigenin used during our study to antagonize AhR (1.65mg/kg) are significantly lower than the doses used in mouse models (20-50mg/kg) (Liang et al. 2017; Yi et al. 2008) or for human clinical trials (up to 500mg capsule 3 times daily of chamomile pharmaceutical grade) (Mao et al. 2016; Shoara et al. 2015) without toxicity. As proposed it is important to optimize the bioavailability and to optimize the doses of apigenin before to be used in clinic in combination with conventional therapies of melanoma (BRAFi/MEKi).

As shown in Figures 1 and 2, even if the flavonoids share very similar structures, their antagonistic function on AhR is different. The antagonist role of Apigenin must be linked to its structure and its ability to interact in the PAS-B domain of AhR. We think that it is important to conserve these chemical properties. To improve the efficacy of apigenin, various steps to optimize bioavailability and prevent rapid degradation would indeed be interesting.

Several teams seek to optimize the effect of flavonoids to improve its solubility, including different delivery systems (liposomes, polymeric micelles, nanosuspension, and so on) (Zhai et al. 2013). The use of optimization protocols could make it possible to improve the effect of apigenin in the context of metastatic melanoma.

These points are now raised in the discussion.  

There are a few typos and missing numbers. I quick read through should fix that.

The different typology errors or missing numbers have been corrected in the text and in the supplemental information.

References

Liang H, Sonego S, Gyengesi E, Rangel A, Niedermayer G, Karl T, et al. OP-25 - Anti-Inflammatory and Neuroprotective Effect of Apigenin: Studies in the GFAP-IL6 Mouse Model of Chronic Neuroinflammation. Free Radic. Biol. Med. [Internet]. 2017;108:S10. Available from: http://www.sciencedirect.com/science/article/pii/S0891584917302678

Mao JJ, Xie SX, Keefe JR, Soeller I, Li QS, Amsterdam JD. Long-term chamomile (Matricaria chamomilla L.) treatment for generalized anxiety disorder: A randomized clinical trial. Phytomedicine [Internet]. 2016;23(14):1735–42. Available from: http://www.sciencedirect.com/science/article/pii/S094471131630188X

Shoara R, Hashempur MH, Ashraf A, Salehi A, Dehshahri S, Habibagahi Z. Efficacy and safety of topical Matricaria chamomilla L. (chamomile) oil for knee osteoarthritis: A randomized controlled clinical trial. Complement. Ther. Clin. Pract. [Internet]. 2015;21(3):181–7. Available from: http://www.sciencedirect.com/science/article/pii/S1744388115000493

Yi L-T, Li J-M, Li Y-C, Pan Y, Xu Q, Kong L-D. Antidepressant-like behavioral and neurochemical effects of the citrus-associated chemical apigenin. Life Sci. [Internet]. 2008;82(13):741–51. Available from: http://www.sciencedirect.com/science/article/pii/S0024320508000374

Zhai Y, Guo S, Liu C, Yang C, Dou J, Li L, et al. Preparation and in vitro evaluation of apigenin-loaded polymeric micelles. Colloids Surfaces A Physicochem. Eng. Asp. [Internet]. 2013;429:24–30. Available from: http://www.sciencedirect.com/science/article/pii/S0927775713002562

Reviewer 2 Report

The authors have done a fine job of attempting to evaluate protective effects of flavonoids in the prevention of AhR-dependent resistance in combination with BRAFi. The hypothesis of this study is interesting.  

I have a number of comments :

Some descriptions of data results are not suitable for introduction section.

It should be clearly addressed rationale of the application of flavonoids in AhR-involved melanoma therapy.

It would be very important to show effects of flavonoids on cell morphology and mitochondria with ultrastructure techniques! It is not sufficient to only demonstrate PCR and qualitative functional results. 

Discussion does not adequately discuss obtained results and needs to focus more on the novelty and clinical impacts of melanoma-AhR-flavonoids axis.

Author Response

The authors have done a fine job of attempting to evaluate protective effects of flavonoids in the prevention of AhR-dependent resistance in combination with BRAFi. The hypothesis of this study is interesting.  

I have a number of comments :

Some descriptions of data results are not suitable for introduction section.

We thank the reviewer for this remark. The end of the introduction has been modified to remove the description of the results appearing too early

It should be clearly addressed rationale of the application of flavonoids in AhR-involved melanoma therapy.

Following this comment, we modified the discussion to clarify the potential therapeutic value of apigenin for melanoma patient treated with conventional therapies (BRAFi / MEKi). Of course, this preliminary study must be completed before claiming a clinical trial, in particular to increase the bioavailability of Api as indicated in the discussion of the article. However, such clinical trial can easily be anticipated by using Api as a nutraceutical formulation in complementation with BRAFi/MEKi, in particular because of the availability of Api, its non-toxicity even at high doses and its beneficial anti-cancer effects. Thus, we proposed in the discussion that Api could be proposed throughout the treatment by BRAFi/MEKi as a food supplements or as a nutraceutical formulation in order to increase BRAFi/MEKi sensitivity and to prevent melanoma relapses. Comparable clinical trials have been already performed to test the efficacy of Genistein and Quercetin (100-500mg / day) in prostate cancer (NCT01538316), with Luteolin vs nano-luteolin in tongue carcinoma (NCT03288298) or using Apigenin in chronic insomnia (NCT01286324) (Zick et al. 2011) or knee osteoarthritis (Shoara et al. 2015).

It would be very important to show effects of flavonoids on cell morphology and mitochondria with ultrastructure techniques! It is not sufficient to only demonstrate PCR and qualitative functional results. 

We thank the reviewer for this comment. The aim of this work was to study on a molecular aspect the role of flavonoids as AhR antagonists and their role on sensitivity to BRAFi. We were able to show that at low doses apigenin was able to inhibit AhR and thus prevent the expression program associated with resistance.

In the course of this study, we did not observe major morphological differences after treatment of melanoma cells with apigenin at the concentrations used (5 μM) (Fig 3b). Nonetheless, we have started to investigate deeply molecular and cellular changes which may operate in response to apigenin, by focusing on mitochondrial morphology changes as shown previously (Seydi et al. 2016). This work will be part of a second manuscript.

Discussion does not adequately discuss obtained results and needs to focus more on the novelty and clinical impacts of melanoma-AhR-flavonoids axis.

We thank the reviewer for this comment. We have modified the discussion by shortening it and by addressing more precisely the potential therapeutic and clinical interest of apigenin in the treatment of metastatic melanoma.

References 

Seydi E, Rasekh HR, Salimi A, Pourahmad ZM and J. Selective Toxicity of Apigenin on Cancerous Hepatocytes by Directly Targeting their Mitochondria [Internet]. Anticancer. Agents Med. Chem. 2016. p. 1576–86. Available from: http://www.eurekaselect.com/node/141448/article

Shoara R, Hashempur MH, Ashraf A, Salehi A, Dehshahri S, Habibagahi Z. Efficacy and safety of topical Matricaria chamomilla L. (chamomile) oil for knee osteoarthritis: A randomized controlled clinical trial. Complement. Ther. Clin. Pract. [Internet]. 2015;21(3):181–7. Available from: http://www.sciencedirect.com/science/article/pii/S1744388115000493

Zick SM, Wright BD, Sen A, Arnedt JT. Preliminary examination of the efficacy and safety of a standardized chamomile extract for chronic primary insomnia: A randomized placebo-controlled pilot study. BMC Complement. Altern. Med. [Internet]. 2011;11(1):78. Available from: https://doi.org/10.1186/1472-6882-11-78